# Coaxial Thermocouples for Heat Transfer Measurements in Long-Duration High Enthalpy Flows

**DOI:** 10.3390/s20185254

**Published:** 2020-09-14

**Authors:** Shizhong Zhang, Qiu Wang, Jinping Li, Xiaoyuan Zhang, Hong Chen

**Affiliations:** 1State Key Laboratory of High Temperature Gas Dynamics, Institute of Mechanics, Chinese Academy of Sciences, Beijing 100190, China; zhangshizhong@imech.ac.cn (S.Z.); wangqiu@imech.ac.cn (Q.W.); zhangxiaoyuan@imech.ac.cn (X.Z.); hongchen@imech.ac.cn (H.C.); 2School of Engineering Science, University of Chinese Academy of Sciences, Beijing 100049, China

**Keywords:** heat transfer measurement, coaxial thermocouple, long-duration, semi-infinite, high enthalpy

## Abstract

Coaxial thermocouples have the advantages of fast response and good durability. They are widely used for heat transfer measurements in transient facilities, and researchers have also considered their use for long-duration heat transfer measurements. However, the model thickness, transverse heat transfer, and changes in the physical parameters of the materials with increasing temperature influence the accuracy of heat transfer measurements. A numerical analysis of coaxial thermocouples is conducted to determine the above influences on the measurement deviation. The minimum deviation is obtained if the thermal effusivity of chromel that changes with the surface temperature is used to derive the heat flux from the surface temperature. The deviation of the heat flux is less than 5.5% when the Fourier number is smaller than 0.255 and 10% when the Fourier number is smaller than 0.520. The results provide guidance for the design of test models and coaxial thermocouples in long-duration heat transfer measurements. The numerical calculation results are verified by a laser radiation heating experiment, and heat transfer measurements using coaxial thermocouples in an arc tunnel with a test time of several seconds are performed.

## 1. Introduction

The accurate prediction of aerodynamic heating is important in the design and development of hypersonic flight vehicles. The peak heating rate in the combustion chamber of a scramjet engine is also a significant parameter in the thermal, structural design of the engine [1]. However, aerodynamic heating prediction remains a difficult problem in modern computational fluid dynamics. Due to the high cost of flight tests, most aerodynamic heating experiments are conducted in ground facilities. Generally, the heat flux data are obtained using temperature sensors that are flush-mounted in the wall of the test model. The time-resolved data are then processed to calculate the heat flux using a physical heat conduction model with few and simplified assumptions. Different ground test facilities have different measurement environments and different test periods; thus, the requirements for the heat flux sensor are variable [2,3]. Transient heat transfer measurements with a test time of milliseconds (i.e., a pulse shock tunnel environment) require a fast response of the heat flux sensor. Commonly used transient sensors include thin-film resistance thermometers and fast-response coaxial thermocouples [4]. However, long-duration heat transfer measurements (i.e., continuous tunnel environment) have relatively low requirements of the sensor response, and commonly used long-duration heat flux sensors include Gardon meters [5] and Schmidt–Boelter meters [6]. Therefore, it is crucial to select a heat flux sensor that is suitable for the test environment.

In supersonic combustion experiments in a direct-connected facility, the engine generally runs a few seconds. The flow field calibration of an arc tunnel is also performed within a few seconds [7]. When heat transfer measurements are conducted in this long-duration environment, the model surface temperature will increase significantly. Thus, the sensor needs to be cooled in most cases and generally has a large size with a diameter of more than 20 mm. However, the internal space of the engine in direct-connected facilities limits the application of long-duration heat flux sensors. The cooling system of the sensor also complicates the system design. Flow field calibrations of arc tunnels also require miniaturized heat flux sensors to obtain sufficient spatial resolution. Coaxial thermocouples based on the one-dimensional (1D) semi-infinite heat conduction theory has the advantages of fast response, strong antierosion capacity, and low production cost; the diameter of these sensors is generally 1–2 mm. Thus, these sensors are easy to install due to their small size and are convenient for heat transfer measurements under these conditions.

Coaxial thermocouples are widely used for transient heat transfer measurements, and the test time in the order of milliseconds meets the assumption of 1D semi-infinite heat conduction. Researchers performed numerous investigations to improve the measuring accuracy of coaxial thermocouples. Sanderson [8] and Marineau [9] conducted studies on the structural design and manufacturing method of coaxial thermocouples and found that the sensor response time was related to the structure of the junction. Li [10] performed a numerical analysis of heat transfer, including the junction of the coaxial thermocouple, and observed a two-dimensional heat transfer effect near the junction due to the influence of the insulating layer. Li also investigated the influencing mechanism of the junction size, the thickness of the insulating layer, and the effect of thermal conductivity on the heat transfer measurements. Marineau [9], Buttsworth [11], Mohammed [12], and Chen [13] conducted calibration experiments on the effective thermal effusivity $\;\sqrt{\rho ck}$ of coaxial thermocouples; large differences were observed in the thermal effusivity for different junction grinding processes. Wang [14] researched the impact of different materials on transient heat transfer measurements obtained from coaxial thermocouples; measurement errors of up to 20% were obtained in 100 ms measurement periods. Since many factors influence the accuracy of heat flux measurements, such as the gauge installation, gauge calibration and sensitivity tests, data reduction procedures, and uncertainties, the accuracy of heat transfer measurements obtained from coaxial thermocouples in a transient environment is ±10% for some simple model shape [15], which might be larger for more complex model shape [16].

In view of the advantages of coaxial thermocouples and their increased use in research on transient heat transfer measurements, coaxial thermocouples have also been used for heat transfer measurement in long-duration facilities (on the order of seconds). Coblish [17] conducted heat transfer measurements on a 25/55° double-cone model using coaxial thermocouples in the No. 9 hypersonic tunnel (HVWT9) at the Arnold Engineering Development Center (AEDC); the effective test time was 15 s. Kirk [18] used coaxial thermocouples for aerodynamic heating measurements on the Orion Crew Module model in the HVWT9 tunnel, and the effective test time was 1 s. Both experiments provided meaningful results, but the complex flow structure or heat flux nonuniformity resulted in challenges in the analysis and use of coaxial thermocouples in long-duration experiments. Additionally, few error analyses were conducted to date of long-duration heat transfer measurements based on the 1D heat conduction theory. Further investigations are also required on the changes in the effective thermal effusivity $\sqrt{\rho ck}$ and the resulting discrepancies in the heat flux at a surface temperature of the thermocouple of several hundred degrees.

In this study, we investigate the use of mature fast-response coaxial thermocouples developed in the laboratory for long-duration heat flux measurements in the order of several seconds to extend the application of the sensors. First, the influencing factors on the long-duration heat transfer measurements obtained from the coaxial thermocouple are analyzed, including the model thickness, the transverse heat transfer, and the changes in the physical parameters of the materials with increasing temperature. Second, the accuracy of the numerical results is verified using a laser radiation heating experiment, and heat flux measurements in the order of several seconds are obtained in an arc tunnel. This study provides theoretical guidance for the design of coaxial thermocouples and the analysis of long-duration heat flux measurements.

## 2. Influencing Factors on the Accuracy of Long-Duration Heat Transfer Measurements Using a Coaxial Thermocouple

### 2.1. Configuration and Principle of Coaxial Thermocouple

The structure of the coaxial thermocouple is shown in Figure 1. A constantan wire of 1.0 mm diameter is inserted coaxially into a machined chromel cylinder of 2.0 mm diameter. The two thermocouple elements are electrically insulated from each other in the radial direction, except at the front surface. The thickness of the insulation is approximate 10 μm. The junction of the sensor is sanded to ensure a smooth surface for the test model. The temperature of the junction is then obtained based on the Seebeck effect. This type-E thermocouple has been widely used in transient heat flux measurement because the thermal properties of the chromel and constantan are similar, thereby reducing the detrimental lateral heat conduction between the two materials. The coaxial thermocouple has the advantages of fast response and good durability.

The surface temperature is measured by the coaxial thermocouple, thus, a mathematical relation is required to derive the heat flux from the temperature. Commonly, it is assumed that the heat conduction inside a surface thermocouple is 1D heat conduction inside a homogeneous semi-infinite solid; thus, two straightforward solutions are obtained [19]:(1)$$T\left(t\right)=\frac{1}{\sqrt{\pi }\sqrt{\rho ck}}\int_{0}^{t}\frac{\dot{q}\;\left(\tau \right)\;}{\sqrt{t-\tau }}d\tau$$
(2)$$\dot{q}\left(t\right)=\frac{\sqrt{\rho ck}}{\sqrt{\pi }}\int_{0}^{t}\frac{dT}{d\tau }\frac{1}{\sqrt{t-\tau }}d\tau$$
where $\dot{q}$ is the surface heat flux; *ρ*, *c*, and *k* are the density, specific heat, and thermal conductivity of the material, respectively; *T* is the measured surface temperature; *t* is the time, and *τ* is the integral variable.

The model thickness and sensor size need to be considered if the semi-infinite assumption is required in the heat transfer measurement. Since the test time in transient heat flux measurements is only on the order of milliseconds, it is easier to meet the semi-infinite assumption. However, if a coaxial thermocouple and Equation (2) are used for long-duration measurements, the assumption of 1D semi-infinite heat conduction will be more challenging.

### 2.2. Effects of Limited Thickness

For an infinite flat plate with limited thickness *l*, the heat flux can be modeled as 1D unsteady-state heat transfer. The temperature in the flat plate can be determined accurately using the 1D unsteady-state differential equation [18]:(3)$$T\left(x,t\right)=T_{0}+\frac{q_{0}l}{k}\left(\frac{\alpha t}{l^{2}}+\frac{1}{3}+\frac{x^{2}}{2l^{2}}-\frac{x}{l}-\frac{2}{\pi^{2}}\overset{\infty }{\underset{n=1}{\sum }}\frac{1}{n^{2}}cos\left(\frac{n\pi x}{l}\right)e^{-\alpha {\left(\frac{n\pi }{l}\right)}^{2}t}\right)$$
where *T(x, t)* is the temperature; *x* is depth in the plate, and *x* = 0 is defined as the surface; *T*_0_ is the initial temperature; *q*_0_ is the applied uniform constant heat flux at the surface of the plate; *l* is the thickness of the plate; *α* is the thermal diffusivity of the material and is defined as:(4)$$\alpha =\frac{k}{\rho c}$$

The heat penetration time *t_p_* provided by Hightower [20] is:(5)$$t_{p}=\frac{l^{2}}{\alpha \pi^{2}}ln\left(2\right)$$

The thermal penetration time *t_p_* can be calculated from the thickness of the plate *l* and the thermal diffusivity *α*. Once the penetration time *t_p_* exceeds a specific value, the semi-infinite assumption is not applicable anymore. Equation (5) can be used to obtain the minimum model thickness required under the assumption of semi-infinite heat conduction for a given test time *t*:(6)$$l>\frac{\pi \sqrt{\alpha t}}{\sqrt{ln\;\left(2\right)\;}}=3.77\sqrt{\alpha t}\approx 4\sqrt{\alpha t}$$

The expression $l=4\sqrt{\alpha t}$ describes the characteristic length under semi-infinite conditions. However, the characteristic lengths vary for different materials due to the differences in thermal diffusivity. In transient heat transfer measurements, the material of the test models is commonly stainless steel, which has an effective thermal effusivity $\sqrt{\rho ck}$ close to that of the sensor material. This material minimizes the influence of the transverse heat transfer between the sensor and the model. The thermophysical parameters of constantan, chromel, and stainless steel at 300 K are listed in Table 1.

The nondimensional Fourier number *F*_0_ is used to obtain universal results for different materials in this study:*F*_0_ = *αt*/*l^2^*(7)

*F*_0_ < 1/16 is obtained by combining Equations (6) and (7), i.e., the 1D semi-infinite heat conduction is satisfied if the Fourier number is less than 1/16 for different plate materials and thicknesses.

The surface temperature of the plate with the thickness *l* is obtained by setting *x* = 0 in Equation (3):(8)$$T\left(t\right)=T_{0}+\frac{q_{0}l}{k}\left(\frac{\alpha t}{l^{2}}+\frac{1}{3}-\frac{2}{\pi^{2}}\overset{\infty }{\underset{n=1}{\sum }}e^{-\alpha {\left(\frac{n\pi }{l}\right)}^{2}t}\right)$$
where *T(t)* is the surface temperature of the plate. The heat flux can be derived from this surface temperature using the 1D semi-infinite heat flux calculation method. Equation (2) can be expressed discretely as follows [17]:(9)$$q=2\sqrt{\frac{\rho ck}{\pi }}\overset{n}{\underset{i=1}{\sum }}\frac{T\left(t_{i}\right)-T\left(t_{i-1}\right)}{\sqrt{t_{n}-t_{i}}+\sqrt{t_{n}-t_{i-1}}}$$

The results of the heat flux versus the Fourier number for various materials are shown in Figure 2. The calculated heat flux is displayed using the nondimensional form of *q/q*_0_, where *q*_0_ represents the heat flux loading at the model surface. Values of *q/q*_0_ closer to one indicate a smaller influence on the measurement results and vice versa. The 1D semi-infinite heat conduction is satisfied when the Fourier number is smaller than 1/16, and the calculated heat flux equals the loaded value. As the Fourier number increases, the deviation between the calculated surface heat flux and the loaded value gradually increases. The deviations are 1% and 10% when the Fourier numbers are 0.255 and 0.520, respectively. Therefore, for long-duration measurements, the thickness of the plate can be increased to obtain a lower Fourier number and smaller measurement deviation.

A test duration of 10 s is considered here, and the thicknesses of the plate need to meet the requirements of the two different Fourier numbers are shown in Figure 3. At 10 s, the characteristic lengths under semi-infinite conditions (*F*_0_ = 1/16) of constantan, chromel, and stainless steel are 31, 28, and 26 mm, respectively. However, in actual experiments, the model thickness is usually limited due to the requirements of the model weight and the strength of the support system. The thickness of the above three materials can be reduced to 15.4, 14, and 13 mm, respectively, when the Fourier number equals 0.255, and the measurement deviation is only 1%.

### 2.3. Effects of Transverse Heat Transfer

The above calculation is based on 1D heat conduction without considering the effects of transverse heat transfer between different materials. Although the thermal effusivity of constantan, chromel, and stainless steel are similar, the effects of the transverse heat transfer still exist. Numerical simulations were conducted to understand the influence of transverse heat transfer on the accuracy of the heat transfer measurements in long-duration experiments. The governing equation is the axisymmetric unsteady heat conduction equation:(10)$$\frac{\partial T\;\left(r,\;z,\;t\right)\;}{\partial t}=\frac{k_{i}}{\rho_{i}c_{i}}\left(\frac{\partial^{2}T}{\partial r^{2}}+\frac{1}{r}\frac{\partial T}{\partial r}+\frac{\partial^{2}T}{\partial z^{2}}\right)\;\left(i=1,\;2,\;3\right)$$
where *r* and *z* are the radial and axial coordinates of the physical space; the other quantities are the same as those in Equations (1) and (2); the subscripts 1, 2, and 3 denote the constantan, chromel, and stainless steel, respectively.

The coaxial thermocouple is simplified to chromel and constantan, ignoring the influence of the insulating layer, which is reasonable because the error caused by the insulating layer will decrease rapidly within a few milliseconds. The details were described in our previous paper on coaxial thermocouples [11]. Inside the sensor and model materials, the temperature and heat flux satisfy the continuity condition at the interface between the two different materials. With the following boundary condition on the top surface
(11)$${\left(\frac{\partial T}{\partial z}\right)}_{z=0}=\frac{q_{0}}{k_{i}}\left(i=1,\;2\right);\;t>0$$
and adiabatic conditions on the other surfaces, Equation (10) is solved using the finite difference method for spatial discretization and the fourth-order Runge–Kutta method for time integration. A code developed in C++ was used in this study; it was verified in reference [10]. The initial temperature is *T*_0_ = 300 K, and a constant heat flux of *q*_0_ = 1.0 MW/m^2^ occurs on the surface. The physical materials parameters used in the calculations are listed in Table 1.

The computational model considered here is assumed to be axisymmetric as shown in Figure 4. The diameter of the sensor is *d*, with a *d*/2 diameter of constantan in the center. The junction is located at half of the sensor radius. Because the main purpose of this calculation is to analyze the influence of the transverse heat transfer between different materials on the heat flux measurement, the contact thermal resistance between different materials is not considered.

Structured grids are applied; the zones near the surface and the sensor/model interface are incorporated with clustered points to provide good spatial resolution. A grid convergence study was conducted for three different grid resolutions, and the thickness *l* = 10 mm was used as an example. There was a negligible difference in the junction heat flux normalized by the loading heat flux for all grids, as shown in Figure 5. Since the junction of the coaxial thermocouple is located between the chromel and constantan, the effusivity in the calculation of the heat flux is the average value of chromel and constantan, i.e., $\sqrt{\rho ck}=\;$8644 (W s^0.5^)/(m^2^ K). Finally, the grid with 400 × 400 grid points was used in the present study.

First, the model thicknesses of *l* = 5 mm and 10 mm are considered here, where the diameter of the coaxial thermocouple is *d* = 2 mm (regular homemade sensors). The calculation time is 10 s. The temperatures at the sensor junction and the heat flux derived from these temperatures using the 1D semi-infinite heat flux calculation method (Equation (9)) are shown in Figure 6. The theoretical temperature obtained from Equation (1) is plotted as well, and $\;\sqrt{\rho ck}$ is the average of the values of chromel and constantan, i.e., 8644 (W∙s^0.5^)/(m^2^∙K). The increase in the surface temperature for the different model thicknesses is consistent with that of the theoretical temperature at the initial time; subsequently, the surface temperature of the sensors deviates from the theoretical value, and the differences increase over time. The error between the calculated heat flux and the loaded value increases over time. At *t* = 10 s, the calculated heat flux values *q/q*_0_ for the two model thicknesses are 1.71, and 1.11, respectively.

The corresponding Fourier number is also shown in Figure 6, where the thermal diffusivity *α* is the average thermal diffusivity of chromel and constantan. If the Fourier number *F*_0_ = 0.255 is considered, as discussed in Section 2.2, the calculated heat flux *q/q*_0_ is 1.03 and 1.04 for the model thicknesses of 5 and 10 mm, i.e., the deviation is 3% and 4%, respectively, and the corresponding time is 1.16 s and 4.65 s. After considering the effects of the transverse heat transfer, the deviation exceeds the calculation result of 1% in Figure 2 under one-dimensional heat conduction.

Different model thicknesses and diameters are considered to obtain universal results and provide guidance for the model design of long-duration tests. The results of the heat flux versus the Fourier number are shown in Figure 7; they cover a wide range of *l/d* from 1.0 to 100. In the calculations, the sensor diameters range from 1 to 2 mm (typical heat flux sensor sizes), and the model thickness ranges from 1 to 200 mm. Compared to the single-material heat conduction with limited thickness, the transverse heat transfer between the sensor and the model material has an influence on the value of *q/q*_0_.

When *l/d* = 1.0, the *q/q*_0_ is very close to 1.0 at the initial moment of *F*_0_ < 0.255. However, when *l/d* = 200, the thickness of the model far exceeds the diameter of the sensor, and *q/q*_0_ is close to 1.05, even at the initial moment F_0_ < 1/16. The reason for this result is that when the thickness far exceeds the diameter, even a smaller Fourier number means a long physical test time, and the surface temperature of the sensor approaches the temperature of stainless steel. However, the effusivity used in the calculation is the average value of chromel and constantan, i.e., 8644 (W∙s^0.5^)/(m^2^∙K), which is 5% higher than the effusivity of stainless steel 8210 (W∙s^0.5^)/ (m^2^∙K). Under other calculation conditions, i.e., when *l/d* is between 1.0 and 100, the curves of *q/q*_0_ are in the gray shaded part of Figure 7. The value of *q/q_0_* increases with an increase in *l/d* for the same Fourier number. As the Fourier number increases, the value of *q/q*_0_ gradually increases, and the trend is similar to the result of the single material.

The effects of transverse heat transfer have to be considered in long-duration aerodynamic heating measurements if high-accuracy measurement results are desired. However, the maximum deviation is less than 5.5% when *F*_0_ < 0.255 and 10.8% when *F*_0_ < 0.520. Therefore, an acceptable measurement deviation can be obtained with proper consideration of *l/d* and the Fourier number, even if transverse heat transfer exists.

### 2.4. Effects of the Sensor Length

The length of the sensor in the above calculation is the same as the thickness of the model. Actually, the length of the sensor is generally longer than the model thickness for convenient installation. The effects of the sensor length on the heat flux measurement in a long-duration test are determined. The calculation model is similar to that in Figure 4, except that the length of the sensor is considered. The length is 20 mm. Similarly, we use the model thickness of *l* = 5 mm and 10 mm as an example; the calculated heat flux *q/q*_0_ for a constant sensor length of 20 mm is shown in Figure 8. For comparison, the results for the same length of the sensor and thicknesses of the model are also shown in Figure 8.

When the model thickness *l* = 5 mm, the *q/q*_0_ decreases from 1.71 to 1.68 as the length of the sensor increases from 5 to 20 mm at *t* = 10 s; this represents a reduction of only 3%. However, when the model thickness *l* = 10 mm, there is no reduction in the value *q/q*_0_ as the sensor length increases. Therefore, an increase in the sensor length has little influence on the calculated heat flux *q/q*_0_.

### 2.5. Effects of the Physical Parameters

In the numerical calculations, the influence of the temperature increase on the thermophysical parameters of the material has not been considered. In transient heat flux measurements, the effect of the temperature increase on the effective thermal effusivity is generally not considered. However, in long-duration heat transfer measurements, the temperature of the sensor and model surface can reach a few hundred degrees if the heat flux is high. In this case, it is necessary to evaluate the effect of the temperature increase on the thermophysical parameters and the measurement accuracy.

The influence of the temperature increase on the thermophysical parameters of type-E thermocouples has been extensively investigated by several researchers [12,22,23]. We used the equation developed by Mohammed [12] to determine the specific heat and thermal conductivity of chromel and constantan with increasing temperature, as well as the thermophysical parameter fitting equation for stainless steel used by Mills [24]. The changes in the effective thermal effusivity versus the temperature of chromel, constantan, and stainless steel are shown in Figure 9. The results are normalized by the effective thermal effusivity at 300 K, as shown in Table 1. The thermal effusivity of the three materials increases with the temperature. The changes in the thermal effusivity of the stainless steel and chromel with increasing temperature are relatively small; however, the effusivity of constantan changes significantly with the temperature. At a material temperature of 600 K, the effective thermal effusivity values of chromel, constantan, and stainless steel are 19%, 50%, and 23% higher than that at 300 K. This result shows the effect of the temperature increase on the effective thermal effusivity of the materials.

Numerical simulations of the change in the thermal effusivity are also conducted to investigate its influence on the heat flux. The numerical model is the same as that described in Section 2.3, and the model thickness remains constant at 10 mm. The surface temperature increase of the junction is different from the results in Section 2.3 (Figure 7) when the changes in the thermophysical parameters are considered, as shown in Figure 10. After the heat flux is loaded for 10 s, the surface temperature increase is 686 K when the physical parameters of chromel, constantan, and stainless steel change with the temperature and 741 K when the physical parameters of the three materials remain constant at 300 K.

The effective thermal effusivity of the sensor is required to derive the heat flux from the temperature increase of the sensor surface. During unsteady heat conduction, the effective thermal effusivity $\sqrt{\rho ck}$ of the model surface changes over time as the temperature increases. It is unreasonable to use a constant thermal effusivity to calculate the heat flux from the temperature. However, the surface temperature of the sensor was measured by the coaxial thermocouple and can be used to calculate the accurate thermophysical parameters at different temperature points. Therefore, Equation (9) can be rewritten as follows:(12)$$q=2\sqrt{\frac{1}{\pi }}\overset{n}{\underset{i=1}{\sum }}\sqrt{{\left(\rho ck\right)}_{T\left(t_{i}\right)}}\frac{T\left(t_{i}\right)-T\left(t_{i-1}\right)}{\sqrt{t_{n}-t_{i}}+\sqrt{t_{n}-t_{i-1}}}$$
where *t_i_* is the time, *T(t_i_)* is the measured temperature at *t_i_*, and $\sqrt{{\left(\rho ck\right)}_{T\left(t_{i}\right)}}$ is the thermal effusivity at *T(t_i_)*.

The primary difference between Equations (12) and (9) is that the influence of the temperature rise on the physical parameters was considered. The constant thermal effusivity in the calculation in Section 2.3 is the average of the effusivity of chromel and constantan at 300 K because their values are similar at this temperature. However, the thermal effusivity of different materials varies significantly with the temperature. Thus, it is critical to choose the appropriate thermal effusivity to calculate the heat flux.

Six different calculation methods were used to derive the heat flux from the temperature increase: the effusivity of the single materials chromel, constantan, and stainless steel; the average effusivity value of chromel and constantan; the average effusivity value of the three materials; and the constant effusivity value of 8644. The results are shown in Figure 11.

The calculated heat flux has a large deviation from the loaded value when the thermal effusivity of the constantan changed with the surface temperature because the change in the thermal effusivity of the constantan varies greatly with the temperature. As a result, regardless of whether the average value of chromel and constantan or the average value of the three materials is used, the calculated heat flux values are significantly different. When the constant value of 8644 is used, the calculated heat flux decreases over time. This method is not suitable for the present analysis, where the thermal physical parameters are changing with the temperature. In contrast, if the temperature-dependent thermal effusivity of chromel or stainless steel is used, the value *q/q*_0_ changes from 1.0 to 1.09 or from 0.95 to 1.05 within 10 s, which means the errors are within 10%. When the parameters of stainless steel are used, the calculated heat flux value is about 5% lower than when the parameters of chromel are used because the thermal effusivity of stainless steel is about 5% lower than that of the chromel. Although the model and sensor temperature fields have some spatial nonuniformity during unsteady heat conduction, the calculated heat flux value *q/q*_0_ has the minimum deviation when the thermal effusivity of the chromel is used after considering the effects of the physical parameter changes with the temperature.

The temperature rise of the model and the sensor under different loading conditions are different, and the changes in the thermal effusivity are related to the temperature. To verify these results, three other heat flux loading values are selected, i.e., 2.0, 0.5, and 0.1 MW/m^2^. The calculated surface temperature and heat flux values for the different heat flux loading conditions are shown in Figure 12.

When the load heat flux is 0.1 MW/m^2^, the maximum temperature of the junction is 343 K, representing an increase of only 43 K. Due to the small temperature increase, the thermophysical parameters of the material change only slightly; hence, the calculated heat flux curve is similar to that when the physical parameters in the calculation do not depend on the temperature (in Section 2.3). When the loaded heat flux is 2.0 MW/m^2^, the maximum temperature is 1010 K, and the thermal effusivity of chromel is 42.6% higher than the parameter at 300 K, as shown in Figure 9. However, the calculated heat flux *q/q*_0_ is similar to the other calculation results under different loading conditions when the thermal effusivity of chromel that changes with the surface temperature is used to derive the heat flux (Equation (12)).

Therefore, after considering the changes in the thermophysical parameters with the temperature, if the temperature-dependent thermal effusivity of the chromel is used, the maximum deviation is less than 4% when *F*_0_ < 0.255 and 10% when *F*_0_ < 0.520 (Figure 12). These results are consistent with those in Section 2.3. The measurement deviation can be minimized if *l/d* is reduced.

## 3. Long-Duration Heat Transfer Measurement Experiments

### 3.1. Laser Radiation Heating Experiment

We use a laser radiation heating method to validate the numerical calculation results and conduct long-duration heat transfer measurements using coaxial thermocouples. The composition of the laser heating system is shown in Figure 13. A high-power laser is used as the energy source. After the laser spot is focused on the integrator through multiple reflections and homogenization, a uniform spot is formed at the exit of the integrator as the heat flux is loaded. A trigger is used to control the laser output time. A power meter is used to determine the output power of the laser. The quality of the laser beam is analyzed with a laser beam measuring instrument to ensure the uniformity of the heat flux loading. An electronic shutter is installed between the test model and the integrator outlet. The shutter is opened after a stable laser output is obtained, and the standard heat flux is applied to the model surface.

The test model is a 50 × 50 mm stainless steel flat plate. The thickness is 10 mm. Three coaxial thermocouples are installed at intervals of 5 mm. The diameter of the coaxial thermocouple is 2 mm, and the length is 20 mm, which are commonly used dimensions of laboratory products. The sensors are labeled as No. 1 to 3. After installing the sensor, the model surface was painted black using a high-temperature black body coating (Pyromark 2500, Tempil Co., New York, NY, USA) to reduce the reflectivity of the surface of the model material. Thus, the model surface is regarded as thermally black, and the absorption coefficient is 0.95. Although the thickness of the paint affects the response time of the thermocouples, the test time in our experiments is long enough to ignore this influence. The heat conduction of the paint was not considered in this study.

The initial temperature of the sensor surface is 302 K, and the time for heat flux loading is 6 s. The surface temperatures of the three coaxial thermocouples are shown in Figure 14a. The curves of the three sensors are very similar, and the maximum temperature is about 470 K after 6 s of heat flux loading. The calculated heat flux curves are presented in Figure 14b. The thermal effusivity of the chromel that changes with the surface temperature is used to derive the heat flux from the temperature, and the heat flux loaded by the laser is 0.55 MW/m^2^. In the first 3 s, the error between the loaded and measured heat flux values is within 3%. The heat flux gradually decreases over time. The reason is that the emissivity of the black body coating material increases with the temperature. The laser radiation heating experiments show that the heat flux calculation using the physical parameters of chromel is accurate and reliable.

### 3.2. Heat Transfer Measurement in an Arc Tunnel

As discussed above, coaxial thermocouples can be used in long-duration heat transfer measurements if an appropriate Fourier number is used. The small size of the coaxial thermocouples facilitates installation. Hence, the sensor can be used flexibly in various heat transfer measurement conditions. In this study, heat transfer measurements are conducted during the calibration of the parameters of an arc tunnel flow field.

Currently, the most common device for cold wall heat transfer measurements in arc tunnels is the copper calorimeter. An air gap or insulating material is generally added between the copper block and the model for insulation. However, local ablation of the sensor surface occurs in experiments, thereby affecting the life of the sensor. Therefore, coaxial thermocouples are used to obtain wall heat transfer measurement in the arc tunnel, and the results are compared with calorimeter measurements.

The test is conducted using arc tunnel equipment with a tubular arc heater, as shown in Figure 15a. The equipment consists of an arc heater, a high-speed nozzle, a test section, and a vacuum system. A two-dimensional rectangular nozzle (120 × 60 mm) is selected to generate high-speed flow (around 2000 m/s). The model is installed close to the nozzle exit. Clean and dry high-pressure air is injected into the arc heater for heating. After the acceleration in the expansion nozzle, the test flow field is formed at the outlet. The model is at a negative angle of attack with the nozzle outlet when the flow field is established so that the model temperature does not increase. When the flow field is stable, the model is quickly adjusted to an angle of attack of 6°, and aerodynamic heating is applied to the surface of the model. A stable flow field is loaded on the surface of the model for about 5 s, and the model is adjusted to a negative angle of attack again to finish the calibration test. The test model is a square stainless steel plate with a size of 100 × 100 mm, a thickness of 10 mm, and 9 sensors. Sensors 1–6 are copper calorimeters, and the insulation material between the copper block and the stainless steel is glass-fiber-reinforced plastic (FRP). C1–C3 are the coaxial thermocouples. The location of the measurement points is presented in Figure 15b, where the airflow is from right to left. The signals from the sensors were acquired by a signal conditioner and were processed on a PC-based data acquisition system at a sampling rate of 100 Hz.

The temperature rise and heat flux values obtained from the coaxial thermocouples are shown in Figure 16. The heat flux values were calculated from the temperature by using the thermal effusivity of the chromel. It is observed that the heat flux decreases from C1 to C3, since they are arranged from front to back in sequence on the plate. The heat flux curves are relatively stable.

On the model, the coaxial thermocouple C2 and the copper calorimeter 3 are the same distance from the front edge of the plate. Thus, a comparison of the measurements obtained from these two sensors is shown in Figure 17. The response time of the coaxial thermocouple is much faster than that of the copper calorimeter; however, the heat flux measured by the copper calorimeter is slightly higher than that by the coaxial thermocouple.

The average value of the heat flux values in 2–5 s are used as the measurements. The measurement results are listed in Table 2. Sixteen-bit AD converters were used in the acquisition board. The overall measurement error of this measurement system was calibrated and was found to be 0.15%.

Table 2 indicates that the heat flux measured by the copper calorimeter is about 10% higher than that measured by the coaxial thermocouple at the same location. The reason is that the FRP thermal insulation material between the copper block and the model that has low thermal conductivity. The surface temperature increase of the FRP is much higher than that of the sensor surface. Therefore, transverse heat transfer occurs on the model surface. The energy is transferred from the FRP to the copper block. Figure 18 shows the surface temperatures of the copper calorimeter. The temperature is highest at the FRP position. The heat energy is transferred from the model surface to the copper block, resulting in high heat flux of the copper calorimeter. Ablation commonly occurs due to the high surface temperature of the FRP, and small particles accumulate and appear in the flow, which results in measurement errors over time. The coaxial thermocouple is thermally matched with the stainless steel material, and there is no significant temperature increase on the surface; thus, there is no risk of local ablation. The results indicate that the coaxial thermocouple provides more accurate and reliable results than the copper calorimeters for heat transfer measurements in the order of several seconds in an arc tunnel.

## 4. Conclusions

A coaxial thermocouple measures the heat flux based on the temperature increase under the assumption of 1D semi-infinite heat conduction. For long-duration heat transfer measurements, it is necessary to consider several influencing factors. In this study, the effects of the finite thickness, the transverse heat transfer, and the physical parameters on the heat transfer measurements are analyzed using a two-dimensional numerical simulation of unsteady heat conduction.

The calculation results indicate that the effects of the changes in the transverse heat transfer and the physical parameters due to increasing temperatures have to be considered in long-duration aerodynamic heating measurements. The deviation of the calculated heat flux q/q_0_ cannot be reduced only by increasing the length of the sensor. The minimum deviation is obtained if the thermal effusivity of the chromel is used to derive the heat flux from the surface temperature. The deviation of the heat flux is less than 5.5% when *F*_0_ < 0.255 and 10% when *F*_0_ < 0.520. This information can be used for the design of test models and coaxial thermocouples in long-duration heat transfer measurement.

The measurement error of the three sensors in the laser radiation heating test is less than 3% in 3 s, which verifies the numerical calculation results and demonstrates the accuracy of the coaxial thermocouples in long-duration heat transfer measurements. For the calibration of heat flux field parameters in the arc tunnel, the heat flux obtained from the coaxial thermocouples is more stable than that obtained from the copper calorimeters. In addition, the response time of the coaxial thermocouples is also faster than that of the copper calorimeter and better describes the physical process of aerodynamic heating.

The present study provides theoretical guidance for the design and analysis of long-duration heat flux measurements using coaxial thermocouples.

## Figures and Tables

**Figure 1 sensors-20-05254-f001:**
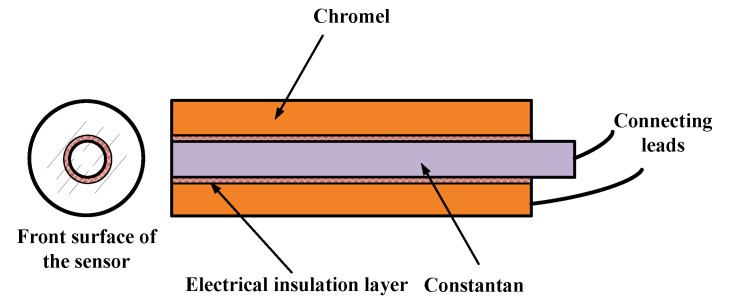
Schematic diagram of the coaxial surface thermocouple.

**Figure 2 sensors-20-05254-f002:**
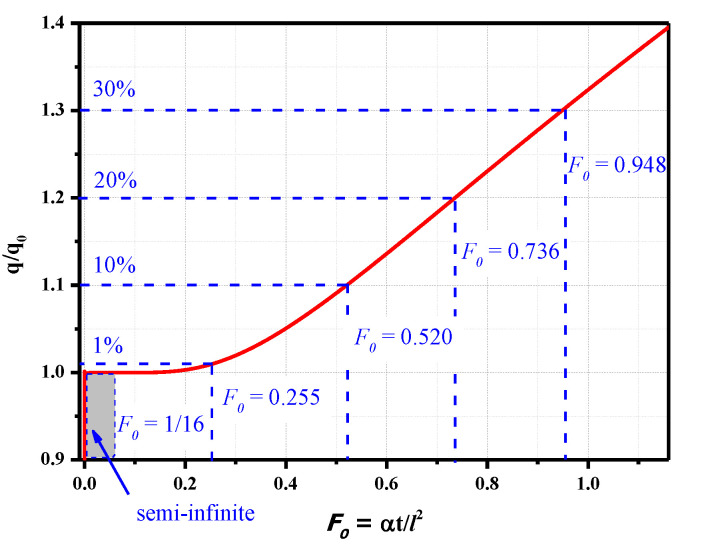
Calculated heat flux versus the Fourier number for 1D heat conduction.

**Figure 3 sensors-20-05254-f003:**
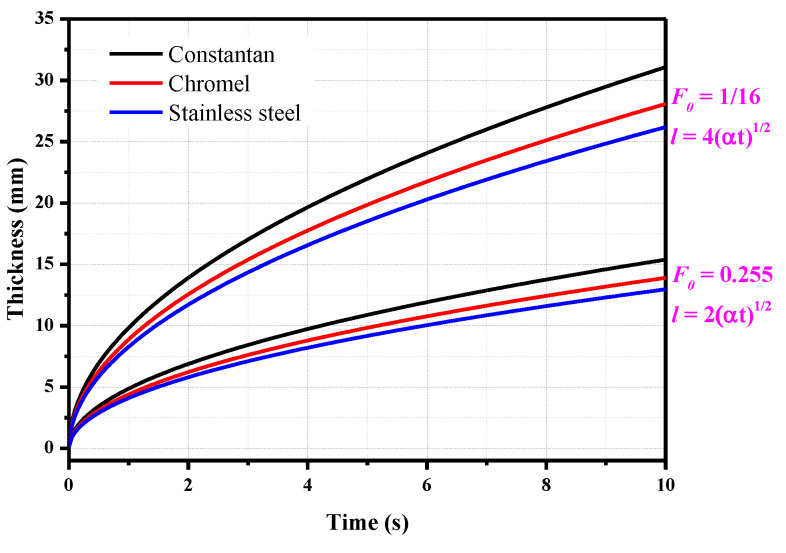
Thickness of the plate versus time for different Fourier numbers.

**Figure 4 sensors-20-05254-f004:**
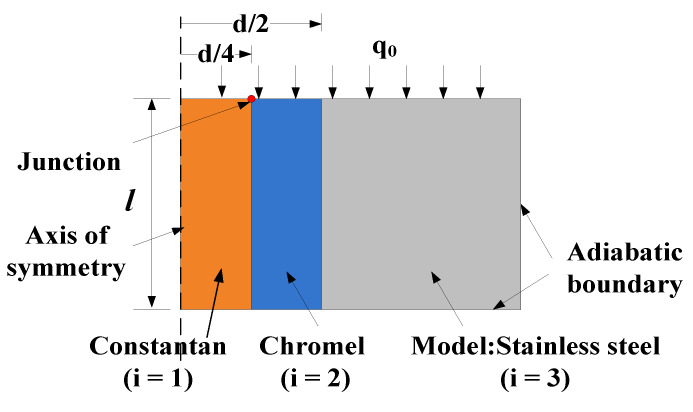
Two-dimensional numerical calculation model (not to scale, units in mm).

**Figure 5 sensors-20-05254-f005:**
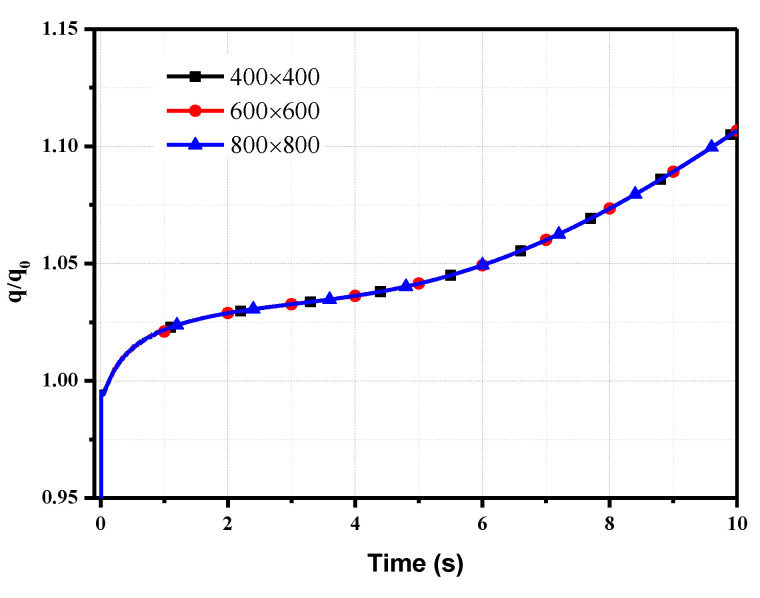
Junction heat flux for three grid resolutions.

**Figure 6 sensors-20-05254-f006:**
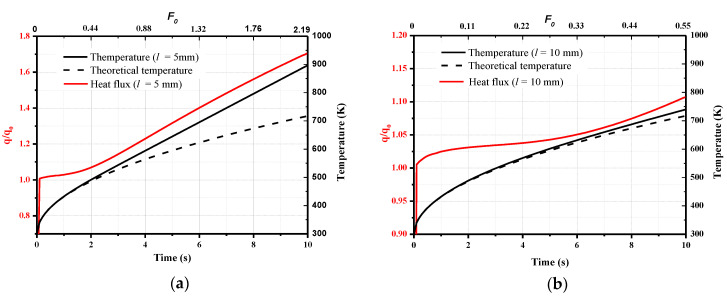
The surface temperature and heat flux: (**a**) *l* = 5 mm; (**b**) *l* = 10 mm.

**Figure 7 sensors-20-05254-f007:**
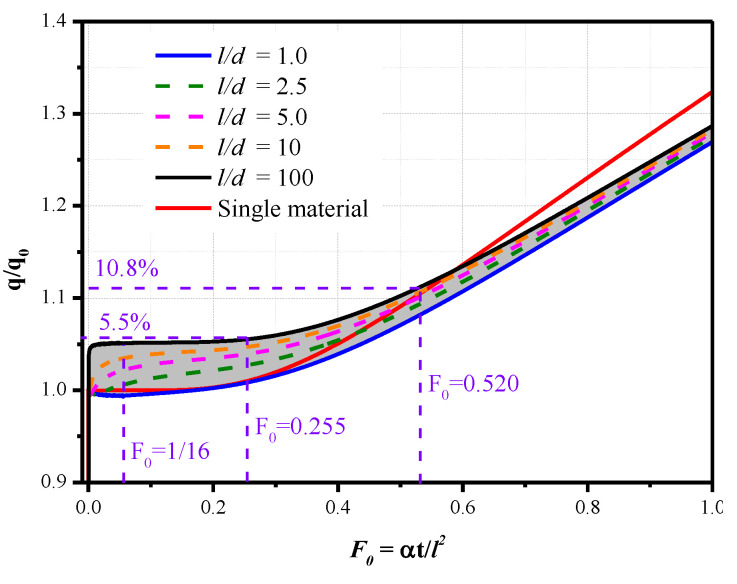
Effects of transverse heat transfer on the calculated heat flux.

**Figure 8 sensors-20-05254-f008:**
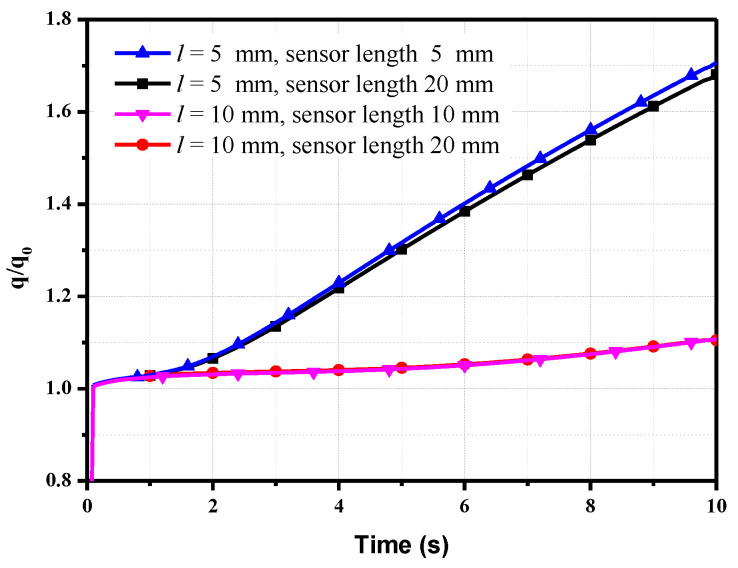
Effects of the sensor length on the heat flux.

**Figure 9 sensors-20-05254-f009:**
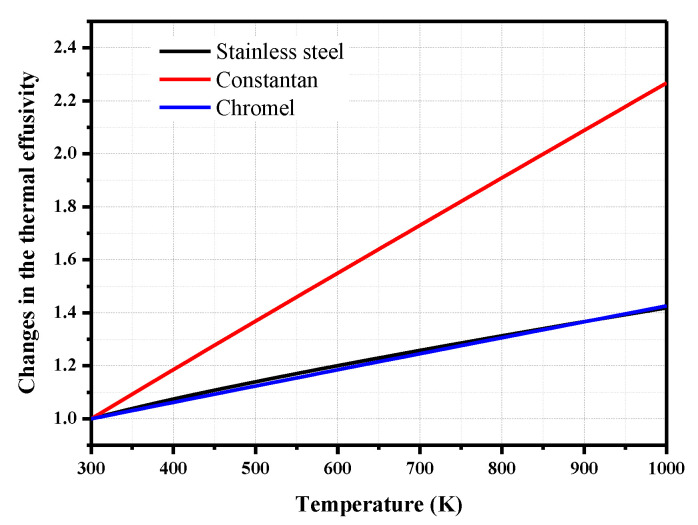
Changes in the thermal effusivity versus the temperature.

**Figure 10 sensors-20-05254-f010:**
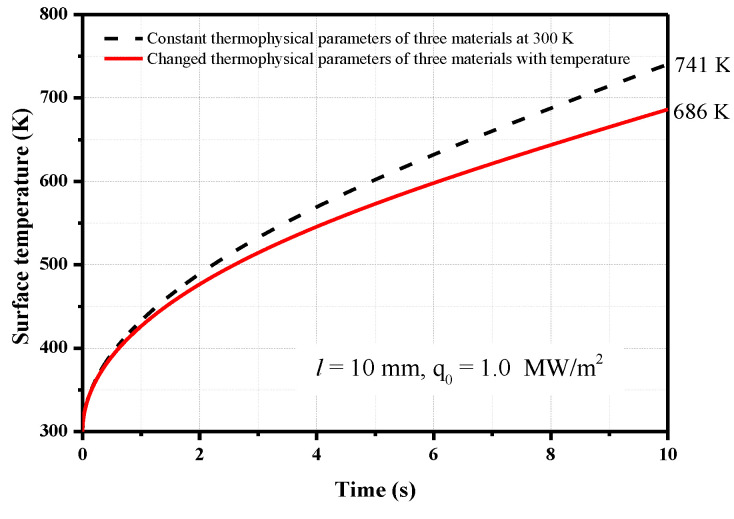
Effect of changes in the thermophysical parameters on the surface temperature of the junction (*q*_0_ = 1.0 MW/m^2^).

**Figure 11 sensors-20-05254-f011:**
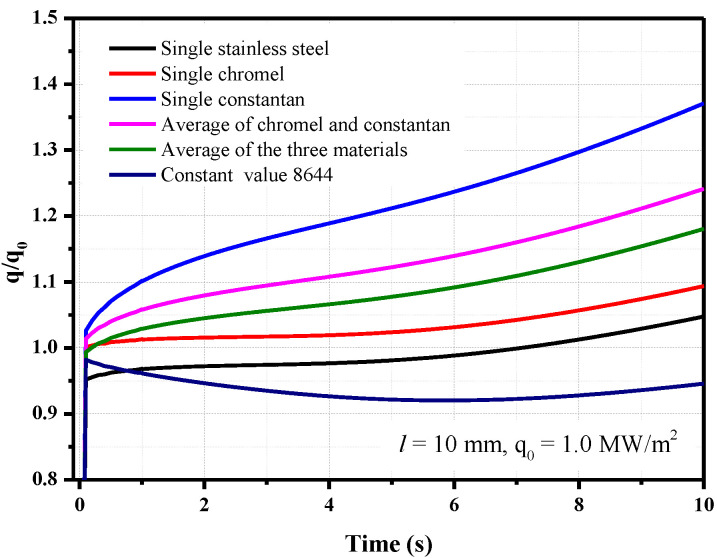
Effect of using different values of the thermal effusivity of the materials on the calculated heat flux (*q*_0_ = 1.0 MW/m^2^).

**Figure 12 sensors-20-05254-f012:**
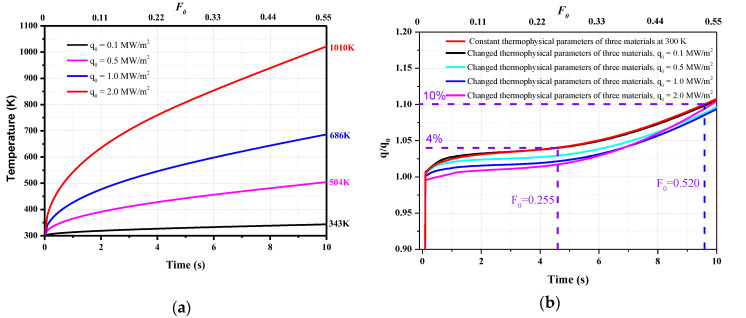
The calculated results for different loaded heat flux values: (**a**) Temperature; (**b**) Calculated heat flux using the thermal effusivity of chromel that changes with the surface temperature.

**Figure 13 sensors-20-05254-f013:**
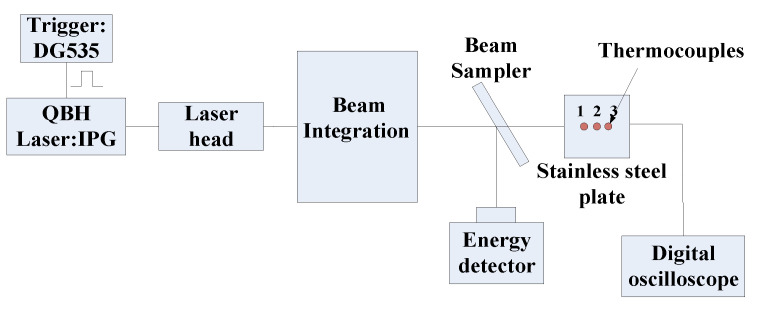
Laser radiation heating system.

**Figure 14 sensors-20-05254-f014:**
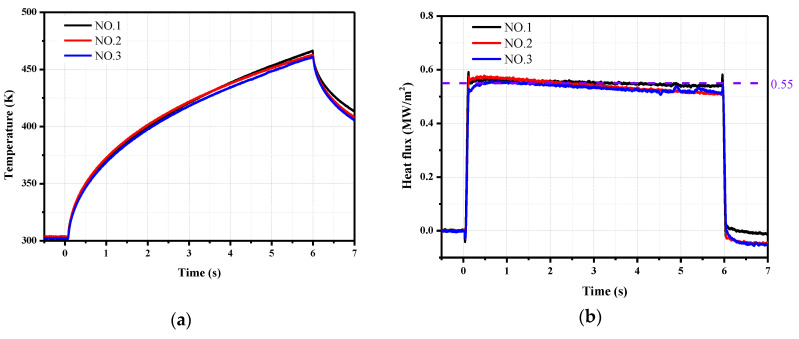
Results of the coaxial thermocouple in the laser radiation heating experiment. (**a**) Surface temperature; (**b**) Heat flux.

**Figure 15 sensors-20-05254-f015:**
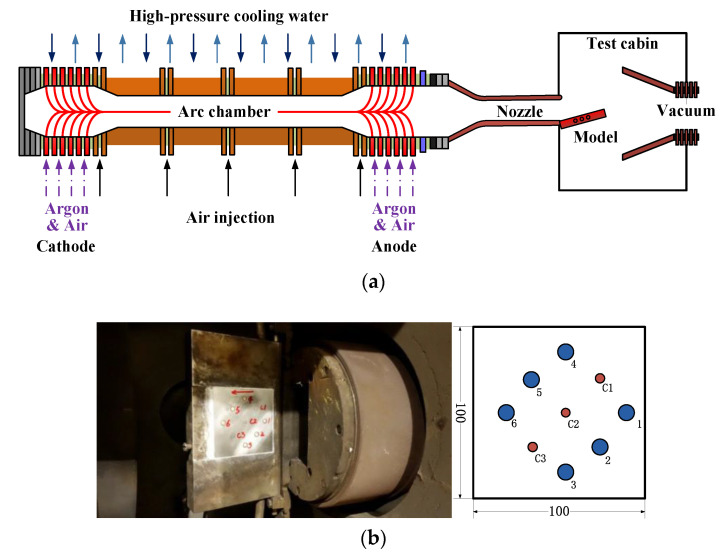
Schematic of the arc tunnel and test model with installed sensors. (**a**) Arc tunnel equipment diagram; (**b**) Mounted position of the sensors on the plate model.

**Figure 16 sensors-20-05254-f016:**
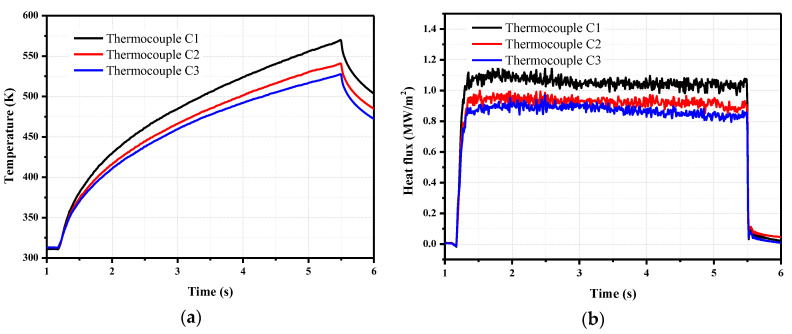
Results of the arc tunnel flow field calibration tests. (**a**) Temperatures of the coaxial thermocouple; (**b**) Heat flux obtained from the coaxial thermocouples.

**Figure 17 sensors-20-05254-f017:**
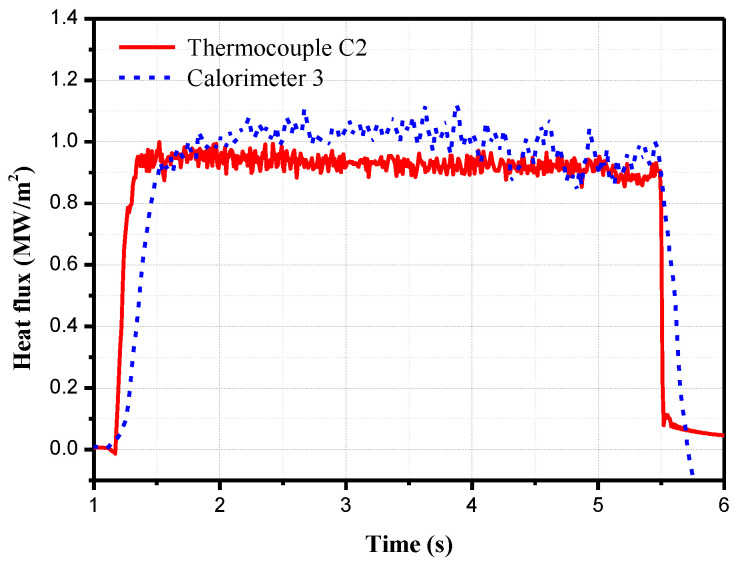
Heat flux obtained from the coaxial thermocouple (C2) and the copper calorimeter (3).

**Figure 18 sensors-20-05254-f018:**
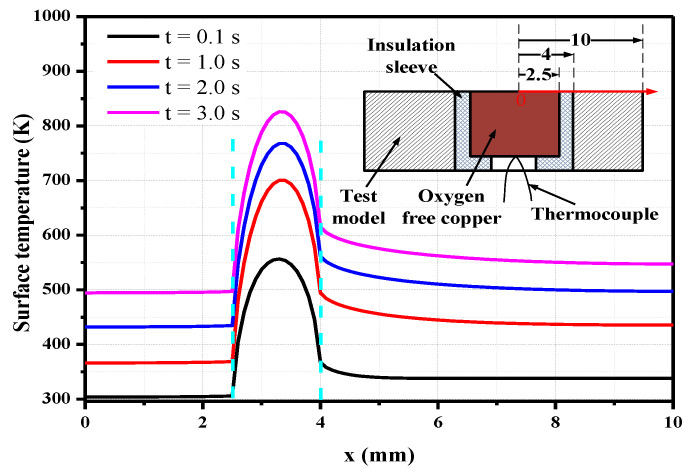
Surface temperature distributions of the copper calorimeter at different test times.

**Table 1 sensors-20-05254-t001:** Thermophysical parameters of the materials at 300 K [21].

Materials	Constantan	Chromel	Stainless Steel
*ρ,* kg/m^3^	8920	8730	7930
*c*, J/(kg·K)	393.1	447.5	500
*k*, W/(m·K)	21.17	19.25	17
α, m^2^/s	$6.04\times 10^{-6}$	$4.93\times 10^{-6}$	$4.29\times 10^{-6}$
(*ρck*)^0.5^, W·s^0.5^/(m^2^·K)	8616	8672	8210

**Table 2 sensors-20-05254-t002:** Comparison of the heat flux obtained from the copper calorimeters and coaxial thermocouples at the same location in the arc tunnel.

Sensor Type	Calorimeter			Thermocouple		
Sensor No.	2	3	5	C1	C2	C3
Measured heat flux (MW/m^2^)	1.166	1.031	0.967	1.051	0.929	0.876

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
