# Peer review of "Coaxial Thermocouples for Heat Transfer Measurements in Long-Duration High Enthalpy Flows"

_sensors, 2020, doi:10.3390/s20185254_

Round 1
Reviewer 1 Report
The authors have overall produced a sound journal publication, however, there are some issues to be addressed and improvements to be made as listed below.
Key to this is the comparison to established methods for determining the suitability of the semi-infinite assumption. I suggest comparing to Schultz and Jones (your ref 17).
There are some modifications that should be made following the laser calibration so that experimentally determined thermophysical values are used.
Based on the method described and some of the comments below I am surprised to see results so closely matched to the calorimeter.
- TITLE- The title should be revised. There is nothing specific to hypersonic flows in this article. Generally, these types of thermocouple are fast-response, but this is not demonstrated by the authors here, nor does it appear critical for their application. Suggested title of "Coaxial thermocouples for heat transfer measurements in high enthalpy flows."
- LINE 10- These thermocouples don't have strong anti-erosion properties. Their durability comes from their ability to be resurfaced.
- LINE 55- does high-resolution data refer to time, space, heat flux, etc?
- LINE 66- Also the Marineau work
- LINE 71-72- can you reference this claimed +/- 10% accuracy? What limits are applied to their operation that makes this so? The statement needs something to back it up.
- LINE 113- the materials are separated in the radial direction, not the axial direction.
- LINE 115- how was 10 micron spacing measured and ensured?
- LINE 129-132- these sentences are unclear and should be re-written. You always need to evaluate the effect of the sensor size when making the semi-infinite assumption. Longer test durations are more challenging for this assumption.
- LINE 148- The sqrt{4 \alpha t} characteristic length is commonly used. Is the above discussion to get to that point necessary>
- LINE 156- The thermal effusivity of these thermocouples is known to be less than the average of the two substrates. This is especially true for fine junctions such as those formed with sandpaper.
- LINE 187- This is a good place to compare your results to those of Schutz and Jones.
- LINE 214- Do you mean modelled as axisymmetric? There's a big difference between axisymmetric and modelling half the model as a plate
- LINE 218- But considering thermal contact conductance would be a much more interesting paper and probably go a long way to explaining why the thermal effusivity of these thermocouples is less than expected.
- LINE 292- You state "the increase in the model surface temperature is usually quite small since the test time is short". This is not true-- You can get surface temperature increases of 100s of degrees C in a millisecond. It's a combination of test time, flow energy and the thermophysical properties of the surface.
- Figure 11- Why single materials? Why consider the SS which will not contribute to the thermocouple sensitivity
- Figure 14- these results are very similar across the three different thermocouples when using the same properties for each. I haven't seen results like this before. How were such tight manufacturing and assembly practices achieved? Why is the heat flux negative initially? Are you applying a constant to convert from emf to T, or are you using a calibration that considers the Seebeck coefficient as a function of temperature?
- Figure 14- the point of these calibration experiments is to determine the thermal effusivity of the sensors... why have you not done this? An experimental value would be much more insightful and useful than the modelled values considering all the assumptions. This must be addressed.
- LINE 421- You comment that cooling is not required because the sensors are small. If these are used without being mounted in a plate, the stagnation heating is much higher for smaller diameter bodies in supersonic/hypersonic flows.
- LINE 433- define high-speed
- LINE 445- If you're sampling at 100 Hz you cannot claim these sensors to be fast response. Especially when others use these with MHz sample rates.
- Figure 16- is the plasma jet heat flux profile such that you expect uniform heat flux across the plate? How have you insulated the front surface of the thermocouple from free electrons in the plasma?
- LINE 459- The sensors cannot be in the same location. If they are, so is calorimeter 4.
- LINE 473- I think the thermophysical properties are more likely a larger contributor to the heat flux differences.
- Figure 18- You stated that epoxy was used to insulate the thermocouple legs. This is certainly melted at the temperatures shown. This is a major concern.
- LINE 495-497- length and material have an influence, but that has been known for a very long time!
- You've captured many key papers in the area which is great.
Reviewer 2 Report
The paper is much improved following expansion of the technical content with further formulae and explanations.
I would still recommend that an English-language revision take place as there are awkward constructions and some grammar errors.
Author Response
Dear Editor and Reviewers:
Thank you for your letter and for the reviewers’ comments concerning our manuscript entitled “Fast-response coaxial thermocouples for heat transfer measurements in long-duration hypersonic flows” (Manuscript ID: sensors-913921). Those comments are all valuable and very helpful for revising and improving our paper. We have studied comments carefully and have made correction which we hope meet with approval. The corrections are highlighted in the file “Marked manuscript”, and the responds to the reviewer’s comments are as flowing:
The paper is much improved following expansion of the technical content with further formulae and explanations.
Thanks for the reviewer’s positive response to our manuscript. We have indeed conducted in-depth investigation on this area for several years and tried to display our work as detailed as possible. The present results are encouraging and valuable for heat transfer measurements by coaxial thermocouples.
I would still recommend that an English-language revision take place as there are awkward constructions and some grammar errors.
Thank you for your suggestions. We have checked the English structure and the grammar in the paper for several times. Moreover, a native English speaker has also been invited to polish the manuscript. We hope the revised manuscript is clear enough to meet the requirements of the reviewer.
Reviewer 3 Report
The contents are interesting. The deliverables are clear. The work seems a worthwhile contribution to the relevant community.
1) To, qo need to be defined right after Eq.(8)
2) Justification for the use of thermal product 8644 needs to be discussed much in detail. The thermal product affects much to the laser calibration data
3) Uncertainty analysis needs to be carried out (Table 2)
4) In 'Conclusions', "... the heat flux ... from the copper calorimeters." needs to be written with care. At the moment, one does not surely know which is more accurate
5) References; the uppercase/lowercase letters mix-up throughout
Author Response
Dear Editor and Reviewers:
Thank you for your letter and for the reviewers’ comments concerning our manuscript entitled “Fast-response coaxial thermocouples for heat transfer measurements in long-duration hypersonic flows” (Manuscript ID: sensors-913921). Those comments are all valuable and very helpful for revising and improving our paper. We have studied comments carefully and have made correction which we hope meet with approval. The corrections are highlighted in the file “Marked manuscript”, and the responds to the reviewer’s comments are as flowing:
The contents are interesting. The deliverables are clear. The work seems a worthwhile contribution to the relevant community.
Thanks for the reviewer’s positive response to our manuscript.
1) To, qo need to be defined right after Eq.(8)
q0 has been used in Eq. (3) for the first time and has been defined right after Eq. (3). Thus, it has not been defined again after Eq. (8).
2) Justification for the use of thermal product 8644 needs to be discussed much in detail. The thermal product affects much to the laser calibration data
You are quite right that the thermal product of the coaxial thermocouple affects much to the heat flux derived from the surface temperature. The influence of thermal product has been discussed in detail in Section 2.5 in this paper. Through analysis, we found that for long duration heat flux measurements, the thermal product changing with temperature must be taken into account, meanwhile, the constant value of 8644 is not suitable as shown in Figure. 11. In view of this, in the following laser radiation heating experiment and arc wind tunnel experiment, the thermal effusivity used is the value of chromel changing with temperature rather than the constant value 8644.
3) Uncertainty analysis needs to be carried out (Table 2)
We agree that it would be better to carry out the uncertainty of the measurement. Thus, we add the information about the influence of the measurement system on the measurement error. A sixteen-bit AD converters were used in the acquisition board. The overall measurement error of this measurement system was calibrated and was found to be 0.15%. We add explanations in the revised manuscript in page 14 and hope the discussion is sufficient enough to meet the requirements of the reviewer.
4) In 'Conclusions', "... the heat flux ... from the copper calorimeters." needs to be written with care. At the moment, one does not surely know which is more accurate
We totally agree that each kind of heat flux sensors has its advantages and disadvantages. It’s really hard to make the conclusion that one is much better the other one. And the measurement accuracy is closely related to the test conditions. However, from the present experimental results, the results from the calorimeters are affected by the lateral heat transfer caused by FRP. And the result from Figure 17 showed that the results from the coaxial is more stable than the calorimeter. Thus, we only keep the conclusion that the heat flux obtained from the coaxial thermocouples is more stable than that obtained from the copper calorimeters.
We made corrections in the revised manuscript and hope the discussion is sufficient enough to meet the requirements of the reviewer.
5) References; the uppercase/lowercase letters mix-up throughout
We have checked carefully and revised the mix-up letter in the references.
This manuscript is a resubmission of an earlier submission. The following is a list of the peer review reports and author responses from that submission.
Round 1
Reviewer 1 Report
This manuscript addressed fast-response coaxial thermocouples for heat transfer measurements. The study provides an important contribution to measurement of heat flux with high time resolution. However, the proposed presentation does not respect the elementary rules of a scientific writing. Authors should strictly observe rules and etiquette of writing paper. For example, Markers are shown in the Fig. 4, 7, 10, 12, 14 and 15, but if this is the acquired data, there is no basis for the curve being displayed in the figures. In other words, the inflection point may have been decided by their own will, and there is a possibility that an act similar to tampering may have been done. Also, Table 2 is not shown according to writing ethics.
I cannot recommend publication of this manuscript.
Reviewer 2 Report
Although an interesting engineering analysis, at this stage I cannot recommend publication of the paper. There are significant questions which arise, and it is not possible to tell whether they were not addressed, or just not described in the paper.
The most significant issue for me is that the bulk of the paper is based on a numerical model which is described in one line as solving the unsteady heat conduction equation. Was this done using a commercial solver? Or was this solved analytically using self developed tools? What sort of validation was done? Model resolution study? The model involves multiple materials bonded/joined somehow. How was this join taken into account in the model? This will certainly affect the heat transfer between the materials (it may be negligible but needs to be addressed). A detailed description of the model is 100% necessary, especially to justify large differences to commonly accepted results (see more below in the details).
The other big picture aspect of the paper that I feel needs improving is the main goal of the study. The paper seems to focus approximately on a 10mm long sensor and a target of 1MW/m2, but is written to as if the results are more generally applicable. This is a bit challenging as there is no indication if the sensor would work at higher heat fluxes (or more specifically to which heat flux it would work), and although it will work for lower heat fluxes, as shown, the design is no longer optimal here and valuable space in the test model is being taken unnecessarily. This could be addressed in two ways, either (1) pick the one geometry and evaulate it in detail for its application range, uncertainty etc. This would essentially be the presentation of a gauge design and the analysis of it. Or (2) a wider parameter study needs to be presented whereby somebody that wants to use a similar gauge could pick a length or heat flux and use the presented data to make a gauge selection. As it is currently, it has aspects of both without either being detailed or specified.
Some other detail comments:
Introduction
Section - Details of the manufacture of the TCs would also be very important. Especially the junction formation. Later, two different possible methods are described, but no statement is made about what was done here.
Eq. 5 - the sqrt should also encompass the ln(2), the maths has been done correctly, it is just a typographical error
Section 2.2
Line 187 - should this read 400% rather than 300%?
Section - I don't think there is much value in showing the 2mm gauge. It is clearly not suitable in this context and makes it more challenging to see the details of the relevant results - particularly looking at Figure 4.
Section - this section concludes that the commonly used semi-infinite method is incorrect by an order of magnitude. Is this only because of the definition of a 10% error being acceptable? This goes against many other findings and needs to be addressed in much greater detail. This is very significant for the paper. Or is there some other explanation? A better presentation may be looking at the percentage error with time and using this as a design parameter.
Section 2.3
Section - Without knowing how the heat transfer between materials is calculated it is very difficult to evaluate the meaning of these results.
Section - The heat flux should only be calculated at the thermouple junction, correct? This means that it is the temperature at this point which is of interest. However, in Figure 6 it is not possible to see any detail of this area. The figure requires either a supplementary figure with more detail, or to be zoomed in to the area of interest.
Figure 7 - I'm assuming this is at 10s? This should be explicitly stated.
Section - It also needs to be very clear that these results are only applicable for 1MW/m2. They are not relevant for other heat fluxes (other than indicative)
Section 2.4
Figure 8 - Although it won't change the information, normalisation would usually presented such that the base value is 1.0 rather than 0.0. Makes it a bit easier to interpret.
Section - I'm not sure what this section is trying to get across. Essentially it is saying that if the incorrect physical parameters (i.e. not temperature dependent) are used then there is a particular error. And if the correct values are used there is a different error, at times even higher than the baseline. This raises questions about the values used and the modelling underlying the results. The concluding paragraph (lines 298-303) states that under very specific conditions and assumptions some values are ok to use, but it is not clear which. Should constant 300K values be used? Should constant 400K values be used? Should the varying values of [12,20-22] be used? e.g. I think if the paper was targetting a specific gauge design then answering these would be much easier.
Section 3.1
Line 321 - All of the paper until this point has been discussing a 10mm length gauge, however the experiment is done with a 20mm gauge? Why was this? Is the 10mm gauge not suitable?
Line 336 - How was the calibration done? The previous statement was that this is commonly done using fluid bath plunging or water dropping methods, but the method used was not stated.
Section 3.2
Section - It is very difficult to see the setup in Figure 13b well, however it appears that the model is mounted at the exit of a rectangular nozzle, how large is the model relative to the flow? Is a spatially constant heat flux expected?
Section - What do the temperature traces of a TC measurement look like? I think this would add great value to the understanding and are more relevant than the temperature trace of the calorimeter.
Reviewer 3 Report
This is a well-presented and fairly well-written (the authors have done a good job with the English overall although one more proofread by a native speaker would not go amiss) study of coaxial thermocouple usage in hypersonic flows. While I would describe the advances contained therein as incremental, they are useful and I would support publication of this article in the present form, or after perhaps just a light proofread.